

# Modelling of post-monsoon drying in Nepal: implications for landslide hazard

Maximillian Van Wyk de Vries[1,2,3], Sihan Li[4], Katherine Arrell[5], Jeevan Baniya[6], Dipak Basnet[6], Gopi K. Basyal[7], Nyima Dorjee Bhotia[6], Alexander L. Densmore[8], Tek Bahadur Dong[6], Alexandre Dunant[8], Erin L. Harvey[8], Ganesh K. Jimee[7], Mark E. Kincey[9], Katie Oven[5], Sarmila Paudyal[7], Dammar Singh Pujara[7], Anuradha Puri[6], Ram Shrestha[7], Nick J. Rosser[8], and Simon J. Dadson[3]

[1]Department of Geography, University of Cambridge, Cambridge CB2 3EL, UK.
[2]Department of Earth Sciences, University of Cambridge, Cambridge CB3 0EZ, UK.
[3]School of Geography and the Environment, University of Oxford, Oxford OX1 3QY, UK.
[4]Department of Geography, University of Sheffield, Winter St, Sheffield S3 7ND, UK.
[5]Geography and Environmental Sciences, Northumbria University, City Campus, Newcastle upon Tyne, NE1 7RU, UK.
[6]Social Science Baha, 345 Ramchandra Marg, Battisputali, Kathmandu, Nepal.
[7]National Society for Earthquake Technology, Sainbu Bhainsepati Residential Area, Lalitpur 13775, Nepal.
[8]Department of Geography, Durham University, Lower Mountjoy, South Rd, Durham DH1 3LE, UK.
[9]School of Geography, Politic, and Sociology, Newcastle University, Newcastle upon Tyne NE1 7RX, UK.

**Correspondence:** M. Van Wyk de Vries (msv27@cam.ac.uk)

**Abstract.** Soil moisture is a key preconditioning factor influencing hillslope stability and the initiation of landslides. Direct measurements of soil moisture on a large scale are logistically complicated, expensive, and therefore sparse, resulting in large data gaps. In this study, we calibrate a numerical land surface model to improve our representation of post-monsoon soil drying in landslide-prone Nepal. We use a parameter perturbation experiment to identify optimal parameter sets at three field moni-

toring sites and evaluate the performance of those optimal parameter sets at each location. This process enables the calibration of key soil hydraulic parameters, in particular a higher hydraulic conductivity and a lower saturation moisture content relative to the default parameter setting. Runs with the calibrated model parameters provide a substantially more accurate (50% or greater reduction in root mean squared error) soil moisture record than those with the default model parameters, even when calibrated from sites as much as 250 km apart. This process enables meaningful calculation of post-monsoon soil moisture

decay at locations with no in situ monitoring, so as to inform a key component of landslide susceptibility mapping in Nepal and other regions where field measurements of soil moisture are limited.

## 1 Introduction

### 1.1 Background

Landslides are a common hazard in Nepal, causing widespread loss of life, population displacement, and damage to infras-

tructure (Adhikari and Tian, 2021; Amatya et al., 2019; Bhandary et al., 2013; Petley et al., 2007). In October 2022, 51 people lost their lives and 778 were affected by landslides across the country, making up almost half of the year's total landslide



fatalities (NSET, 2023). This raises the question of why so many fatal landslides occurred in that particular month. A heavy rainstorm occurred in early October, but its intensity was comparable to several events earlier in the year, and total October rainfall was less than half that of the totals in June, July, or August. This article focuses on another potential contributing fac-

tor: soil moisture. Soil moisture content can play an important role in preconditioning slopes for failure (Bogaard and Greco, 2016; Marino et al., 2020; Ray et al., 2011), and was at its highest point in October immediately after the monsoon. However, direct measurements of soil moisture are rarely available on a spatial or temporal scale relevant for landsliding. In this article, we explore the feasibility of using sparse field observations to calibrate a land surface model and enable more accurate and widespread mapping of soil moisture, with the ultimate objective of improving the accuracy of landslide forecasts in Nepal.

From a mechanical perspective, a landslide is a gravity-driven downslope movement of rock and soil, triggered when the driving forces of a given hillslope overcome its resistive forces (Bogaard and Greco, 2016; Záruba and Mencl, 2014). Despite an improvement in our understanding of the physical mechanisms triggering landslides and the tools available to monitor susceptible areas, skilful prediction of future landslide occurrence remains a major challenge (Casagli et al., 2023; Lacroix et al., 2020; McColl, 2022; Petley et al., 2017). Since the small-scale processes determining the stability of a single hillslope

cannot be measured across an entire mountain range, the susceptibility of a hillslope to failure is commonly divided into predisposing and triggering factors (Carlà et al., 2019; Casagli et al., 2023; McColl, 2022). Predisposing factors are established by the local geological, geomorphological, and hydrological conditions and determine how susceptible a given hillslope is to failure in its baseline state. Triggering factors are external modifications to a hillslope that may precipitate its failure, such as earthquake shaking (Harp and Jibson, 1996; Xu et al., 2016), intense rainfall (Caine, 1980; Dahal and Hasegawa, 2008;

Carey et al., 2019), and fluvial or anthropogenic activity (Cook et al., 2018; Petley et al., 2007; Yang et al., 2021). Predisposing and triggering factors can be estimated using datasets more readily available on a large scale, such as surface topography, and enable large-scale landslide susceptibility mapping.

Soil moisture is a key predisposing factor, as it can simultaneously reduce a hillslope's resistive forces through an increase in pore water pressure and reduced internal friction and increase a hillslope's driving forces through increased bulk density

(Bogaard and Greco, 2016; Marino et al., 2020; Ray et al., 2011; Wicki et al., 2020). Changes in soil moisture are also closely associated with rainfall, which is one of the most widespread landslide triggers (e.g., Caine, 1980; Dahal and Hasegawa, 2008; Kirschbaum and Stanley, 2018). Therefore, as volumetric soil moisture decreases following a large rainfall event or wet period, we expect a change in landslide susceptibility. However, while the theory linking soil moisture to landslides is well established (e.g., Bogaard and Greco, 2016), translating this into practice for operational landslide forecasting remains challenging (Marino

et al., 2020; Ray et al., 2011; Wicki et al., 2020), in part due to the complexity of assessing soil moisture conditions on a large scale.

Soil moisture can be measured directly in the field, with the most common sensor type calculating moisture content by measuring soil capacitance (Eller and Denoth, 1996; Placidi et al., 2020). While the accuracy of these sensors is high in ideal conditions (within a few %), large variations in soil moisture over small spatial scales complicate their extrapolation to a

larger area. More complex field monitoring methods for soil moisture have been developed based on cosmic ray neutron flux, which has the advantage of a larger monitoring footprint (Andreasen et al., 2016; Baatz et al., 2014; Zreda et al., 2008, 2012).



Satellite monitoring has been used to enable regional to global-scale monitoring of soil moisture, including active (radar), passive (radiometer), and combined active-passive measurements (Chan et al., 2018; Colliander et al., 2017; Dorigo et al., 2017; Entekhabi et al., 2010). Satellite-based soil moisture products and their derivatives generally have a coarse spatial resolution ( $10^1$-$10^2$ km) and only measure moisture in the uppermost soil layer (within ∼5 cm of the surface), leading to high uncertainties in certain areas (Chan et al., 2018; Dorigo et al., 2015, 2017).

An alternative or complementary approach to soil moisture monitoring is to model the infiltration and movement of water within the soil. Soil moisture evolution models typically calculate a numerical solution to the Richards equation (Beven, 2012; Richards, 1931, 1949) for the vertical flow of liquids through a porous medium coupled with a range of possible complementary modules. Typical additional modules are a vegetation and evaporative flux model to evaluate moisture losses at the soil surface, and a full climate model or groundwater flow model can also be coupled to the soil model under certain circumstances. Soil moisture models allow for the calculation of soil moisture from meteorological data, which are more readily available on a large scale, but remain highly reliant on appropriate local parameter choices.

## 1.2 Objectives

A fundamental research problem in landslide science is understanding which slopes are most susceptible to landsliding before collapses occur. Where possible, this is further extended to include a specific prediction of the time of hillslope failure. In reality, this remains a major challenge even at the best monitored and instrumented field sites, let alone regions with sparse or no in-situ data (Bogaard and Greco, 2016; Caine, 1980; Casagli et al., 2023; Glade, 2001; Hilker et al., 2009; Marino et al., 2020; McColl, 2022; Petley, 2012; van Westen et al., 2006; Záruba and Mencl, 2014). The mechanical link between water content and hillslope (in)stability is well-established (e.g Bogaard and Greco, 2016), and both precipitation and soil moisture are used to improve landslide susceptibility maps and produce estimates of future landsliding (Bogaard and Greco, 2016; Caine, 1980; Kirschbaum and Stanley, 2018; Marc et al., 2018; Marino et al., 2020; Ray et al., 2011; Wicki et al., 2020).

In this study, we use a network of field monitoring stations in Nepal to calibrate the Joint UK Land Environment Simulator (JULES) land surface model for improved representation of post-monsoon soil drying (Best et al., 2011; Clark et al., 2011; Cooper et al., 2021). We use a parameter perturbation experiment across three different sites and four precipitation datasets to identify a locally optimal parameter set for a pedotransfer function and evaluated this by cross-comparison between geographically distant sites. We then consider the implications of more accurate soil moisture model outputs for landslide occurrence and hazard assessment in Nepal. This study provides a mechanism for upscaling high-precision point measurements of volumetric soil moisture from field monitoring stations for regional landslide hazard assessments. This provides several advantages over satellite-based monitoring (Kirschbaum and Stanley, 2018; Zhao et al., 2021) as JULES can be run at a higher resolution over complex terrain, provides information on moisture content in both surface and deep soil layers, and can be driven with weather forecast and climate projection data to estimate future land surface hydraulic conditions (Best et al., 2011; Clark et al., 2011; Cooper et al., 2021).



## 1.3 Study Locations

85 Nepal, a mountainous nation located at the southern margin of the Himalaya, is susceptible to landslides for multiple reasons. First, Nepal is home to extreme topographic gradients, with the second-largest elevation range of any country on Earth and steep topography over much of the country. Second, it experiences very high rainfall frequency and intensity, particularly during the Indian summer monsoon (herein referred to as 'the monsoon') from June to September (Ichiyanagi et al., 2007). Finally, Nepal is seismically active due to the presence of several large faults along the Indian-Eurasian plate boundary. This high landslide 90 hazard intersects with high landslide exposure, with many of the steep slopes also having high densities of population and key infrastructure. Better understanding and mitigation of landslide hazard is crucial for disaster risk reduction and preparedness in Nepal.

Nepal's strong seasonality in precipitation translates into a strong seasonality in soil moisture content, with soils commonly approaching saturation during the monsoon and then drying down to their lowest point in December-February. During the 95 post-monsoon period of September to December, soils dry from near-complete saturation to their lowest soil moisture content (Ichiyanagi et al., 2007; Nepal et al., 2021; Talchabhadel et al., 2019). The reduction in soil moisture content over these 2-3 months is expected to also manifest as a reduction in landslide hazard.

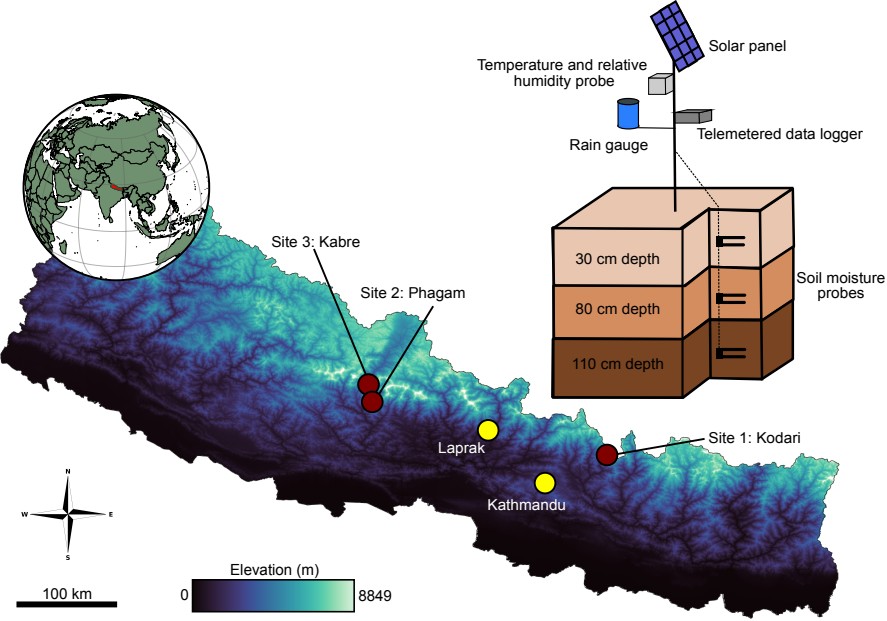

**Figure 1.** Location of the field monitoring sites in Nepal, alongside a diagram of the key components of each monitoring station.

Here, we use field data from three different sites across the mountain and hill districts of Nepal. The first site is located at Kodari in Sindhupalchok district, a few kilometres from the Nepal-China border and at an altitude of 1948 m (Figure 1). The 100 second and third sites are at Phagam and Kabre in Myagdi district at altitudes of 1493 m and 1699 m respectively, around 250





km to the west of the first station (Figure 1). This separation between the different sites of our monitoring network allows us to test the applicability of a calibrated parameter set beyond the immediate local area in geographically, topographically, and climatically distinct areas. For instance, the site at Kodari is located in a wetter part of Nepal than those at Phagam and Kabre (e.g. Figure 2).

## 2 Methods

### 2.1 Field monitoring sites

Each field monitoring station is composed of three main components: a basic weather station, a network of soil moisture probes, and a telemetered data logger. The weather station comprises a tipping-bucket rain gauge for measuring liquid precipitation and sensors measuring temperature and relative humidity (Figure 1). This setup cannot accurately measure solid precipitation (snow or hail), which is rare but possible in these areas. The soil moisture probe network consists of three Tempcon S-SMD-M005 Soil Moisture Smart Sensors, which use 10 cm long capacitive probes to measure volumetric soil moisture. The stated accuracy of these probes is $\pm$ 3% for temperatures within the range 273-323 K (0 to 50 °C), which covers the usual annual temperature cycle at each of our study sites. All three probes were installed in September 2022 at depths of 30 cm, 80 cm, and 110 cm to monitor changes in soil moisture with depth (Figure 1). A HOBO RX3000 logs the data and transmits it over the local 4G network every hour. A battery powers the station and is recharged during the day by a solar panel mounted on the top of the mast.

### 2.2 Parameter perturbation experiment workflow

In this paper we use JULES, a Land Surface Model (LSM), to model changes in soil moisture (Best et al., 2011; Clark et al., 2011). As with most LSMs, JULES takes a standard suite of meteorological variables, including temperature, precipitation, radiation, humidity, and wind speed, as driving data. We use a LSM as the changes in moisture content are closely tied to changes in vegetation, snowpack, and surface energy fluxes and cannot be modelled in isolation (Best et al., 2011; Clark et al., 2011; Cooper et al., 2021). JULES models changes in soil moisture using the Richards equation (Beven, 2012; Richards, 1931, 1949). JULES is integrated into the UK Met Office Unified Model and has been extensively evaluated across a wide range of environments.

To model soil moisture, JULES requires information about the soil hydraulic conditions, such as the soil hydraulic conductivity, soil matric suction, and saturation volumetric water content. Information about these hydraulic parameters themselves is rarely available, so they are commonly related to more easily measurable soil physical characteristics, such as the grain size distribution of minerals within the soil, through a pedotransfer function. Here, we use the empirical and continuous Cosby pedotransfer function (Cosby et al., 1984), which relates a soil's sand, clay, and silt content to measurable hydraulic parameters. We select the Cosby pedotransfer functions over alternative formulations for three main reasons: (i) they have been widely used and evaluated across a range of environments around the globe; (ii) they are parsimonious, requiring only easily-accessible in-



formation about sand, silt, and clay content instead of more complex characteristics such as soil density and pH; and (iii) they can be used to calculate each soil hydraulic parameter required to run JULES. The specific form of these pedotransfer functions are:

$$b = \kappa_1 + \kappa_2 f_c + \kappa_3 f_{sa} \tag{1}$$

$$\theta_S = \kappa_4 + \kappa_5 f_c + \kappa_6 f_{sa} \tag{2}$$

$$\Psi_S = 10^{\kappa_7 - \kappa_8 f_c - \kappa_9 f_{sa} - 2} \tag{3}$$

$$\Psi_S = \frac{127}{180} 10^{\kappa_{10} - \kappa_{11} f_c - \kappa_{12} f_{sa} - 2} \tag{4}$$

with b being the soil hydraulic characteristic exponent in the Brooks and Corey soil physics description, $\theta_S$ being the volumetric water content at saturation, $\Psi_S$ being the soil matric suction at saturation, $K_S$ being the soil hydraulic conductivity at saturation, $f_c$ being the clay fraction, $f_{sa}$ being the sand fraction, and $\kappa_1 - \kappa_{12}$ being the pedotransfer calibration constants. These equations can be further used to calculate the soil moisture at the critical ($\theta_{crit}$) and wilting point ($\theta_{wilt}$), and the dry soil heat capacity ($h_{cap}$) and thermal conductivity ($h_{con}$) if information about the silt fraction ($f_{si}$) is available. To enable wider applicability, we use local soil texture data, specifically the local (within a 250 m grid cell) fraction of sand, silt, and clay, from the global SoilGrids 2.0 database (Batjes et al., 2020; Poggio et al., 2021; de Sousa et al., 2020).

JULES requires precipitation driving data to compute soil moisture. Soil moisture is particularly sensitive to precipitation inputs, and any bias or error in precipitation-driving data will propagate into errors in modelled soil moisture content. To ensure the results are applicable to a wider geographical area, this driving data must come from a regional or global dataset. Here, we select four distinct precipitation datasets: CHIRPS (Funk et al., 2015), ERA5-land (Muñoz-Sabater et al., 2021; Xu et al., 2022), GPM IMERG (Huffman et al., 2019; Pradhan et al., 2022), and MSWX (Beck et al., 2019, 2022). We obtain the non-precipitation variables from MSWX, which provides all the variables needed to run JULES within a single consistent framework (Beck et al., 2022). ERA5-land, GPM, and MSWX all have a spatial resolution of 0.1 degrees ($\sim$10 km), while CHIRPS has a spatial resolution of 0.05 degrees ($\sim$5 km). The monitoring stations are all located within deep valleys ($\sim$5000 m vertical relief) with a width of 10-20 km, so we may expect the higher spatial resolution of CHIRPS to better capture local precipitation patterns (Funk et al., 2015). The resolution of the data nevertheless remains coarse, and all of these datasets have known limitations over mountainous terrain such as the Himalaya (Beck et al., 2019; Funk et al., 2015; Ma et al., 2018; Pradhan et al., 2022; Xu et al., 2022).

We calibrate an optimal set of Cosby pedotransfer function constants ($\kappa_1 - \kappa_{12}$) instead of the soil hydraulic parameters as they exhibit lesser covariance and non-linearity, and allow for greater transferability of optimal parameter sets between



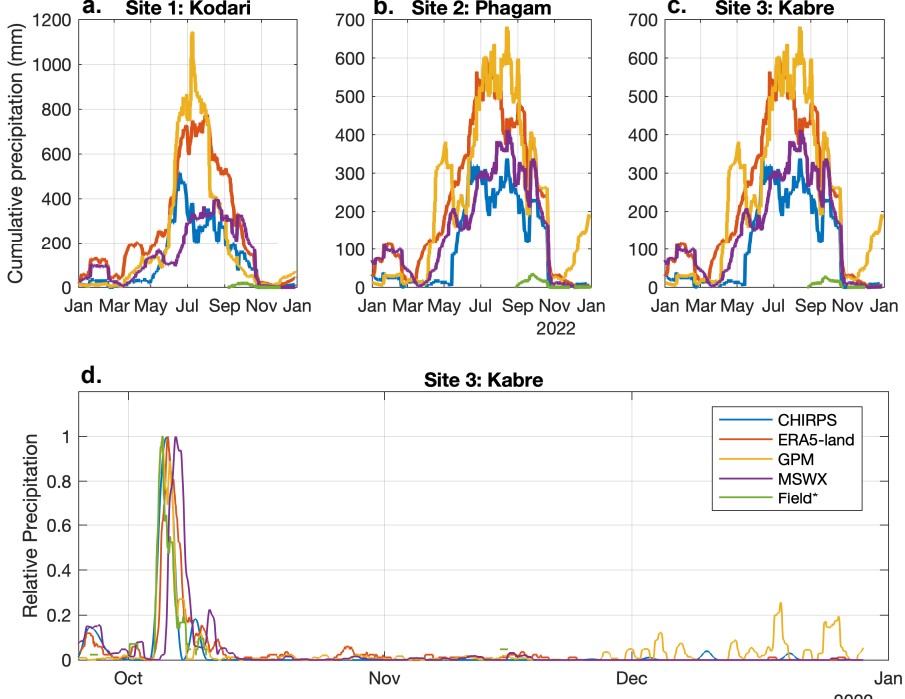

**Figure 2.** Precipitation comparison between the four precipitation sources and field data in 2022. Note the field data was not recorded before September 2022, and therefore does not include any monsoon rainfall. The relative precipitation (precipitation normalized by the highest rainfall value in the time series) patterns between the datasets are similar, but the absolute values differ by as much as a factor of 2.

different sites. We use a Parameter Perturbation Experiment to explore the parameter space of each of the 12 pedotransfer function constants. We set up the same 1000-member ensemble for the 3 different sites and four different precipitation driving datasets for a total of 12,000 ensemble members. We use a Latin hypercube sampling procedure, which ensures that the sampled parameter space is representative of real variability, and adjusts the value of each pedotransfer function constant within $\pm$ 50% of its default (Cooper et al., 2021; Cosby et al., 1984). We run this ensemble on the JASMIN high-performance computing

facility (https://jasmin.ac.uk). To avoid initialization artefacts, we spin up the model for three years and run it for all of 2022 ˘ for a total model time of 4 years.

We compare each model output to the field data using two different metrics. The first metric is the Kling-Gupta efficiency (KGE; Gupta et al., 2009), which compares a combination of the correlation mean bias, and the difference in variability between the two volumetric soil moisture time series. It is computed as:

$$KGE = 1 - \sqrt{(r-1)^2 + (\alpha - 1)^2 + (\beta - 1)^2} \tag{5}$$





with r being the linear correlation between the field and model time series, $\alpha$ being the ratio of the standard deviation of the field and model time series, and $\beta$ being the mean of the field data divided by the mean of the model time series. The KGE score approaches 1 as the similarity between the two time series increases. We linearly interpolate between the JULES model layer depths (5±5 cm, 22.5±12.5 cm, 67.5±32.5 cm, and 200±100 cm) to obtain soil moisture at the exact depths of the field
measurements (30, 80, and 110 cm; Figure 1). The advantage of the KGE is that it combines multiple distinct metrics for the similarity between the two time series and can more effectively describe their similarity than a simple correlation or mean squared error.

The second comparison metric is based on the observation that virtually no precipitation occurs during the October-December post-monsoon drying. The soil is therefore simply drying over this period with little to no increase in moisture content. We,
therefore, fit the soil moisture time series for each soil layer from 18 October to 18 December, which avoids the early October rainstorm, with an exponential decay function (Shellito et al., 2016):

$$\theta(t, z) = \theta_f + Ae^{-\frac{t}{\tau}} \tag{6}$$

with $\theta$ being soil moisture content at a given time t since the beginning of drying and depth z, and $\theta_f$, A, and $\tau$ being fitting parameters used to determine the optimal exponential fit to a given soil moisture time series. We fit an exponential decay to
each time series via nonlinear least squares to determine the optimal fitting parameters $\theta_f$, A, and $\tau$ as well as the coefficient of determination and root mean squared error of each fit. We construct a cost function C to evaluate the overall similarity between the fitting parameters as:

$$C = \frac{|\theta_f^M - \theta_f^O|}{\overline{\theta_f}} + \frac{|A_f^M - A_f^O|}{\overline{A_f}} + \frac{|\tau_f^M - \tau_f^O|}{\overline{\tau_f}} \tag{7}$$

with the M and O superscripts corresponding to the model (single ensemble member) and observation fit parameters, respec-
tively, and $\overline{\theta_f}$, $\overline{A_f}$, and $\overline{\tau_f}$ corresponding to the mean value of each fit parameter across the model ensemble for each site and precipitation dataset.

To determine whether the optimal parameter set from one field site can be used to improve the representation of soil moisture over a larger area, we compare the optimal parameter sets across the different stations. We use either the KGE or cost function to compute an optimal parameter set for each site. We then compare the optimal JULES model runs from the other two sites
to the median of the model ensemble and the mean of the field station data, corresponding to a KGE of -0.41 (Knoben et al., 2019). This KGE value of -0.41 provides a reference point compared to a constant timeseries equal to the data mean. Sites 2 and 3 are around 5 km apart, while Site 1 is around 250 km from Sites 2 and 3, enabling an assessment of the distance over which local parametrisation is valuable.





## 3   Results

### 3.1   Field data

The soil moisture and meteorological variables were recorded at hourly intervals without interruption from the time the stations were installed in mid-September 2022 through 2023 (and continue to record at the time of writing). Battery voltage information confirmed that the stations remained fully powered and functional at all times. The rain gauge records a small amount of precipitation (20-30 mm) in late September and early October, followed by a prolonged dry period with at most three rainy days and <1 mm/day of precipitation at all three sites over the entire period (Figure 3). All three sites show a similar overall pattern of soil moisture over the study period, with slight water recharge during the rainfall events early in the record followed by continuous drying. The shallowest soil layer (30 cm depth) exhibits a weak diurnal cycle and dries most rapidly at each site. At Site 2 (Phagam), the shallowest soil layer exhibits a break in slope in mid-November, after which soil moisture declines to a much lower value than the other two sites (reaching less than 0.1 $m^3.m^{-3}$ soil water content by the end of December, versus 0.2 $m^3.m^{-3}$ and 0.15 $m^3.m^{-3}$ for 1 and 3 respectively). We do not know the exact cause of this rapid drying at Site 2, which may result from soil structure changes due to hillslope instability, agriculture-related water extraction, or sensor malfunction. The second soil layer (80 cm) at Site 3 (Kabre) exhibits two clear step increases in soil moisture, which are not reflected in the shallow (30 cm) or deep (110 cm) layers and occur on days with no precipitation. We subtract these step changes, which aligns the mid-depth soil moisture time series with that of the lower and upper layers (Figure 3), but treat this record with caution in all further analyses.

### 3.2   Results of the parameter perturbation experiment

The parameter perturbation experiment allows us to determine which model parameters and driving datasets provide the closest match to field data. All ensemble members show the same broad seasonal pattern in soil moisture content, driven by the seasonal cycle in precipitation (Figure 4). Mean volumetric soil moisture reaches a maximum early in the monsoon (June) and high values are maintained until September, after which moisture steadily declines until the end of the year. CHIRPS-driven ensemble members exhibit the steadiest decline after the monsoon, while ERA5-land and MSWX exhibit spring (February-March) peaks in soil moisture due to early precipitation events. The deeper soil layers (80 cm and 110 cm) fully saturate during the monsoon for model runs driven by ERA5-land or GPM, which exhibit higher total annual precipitation values (Figure 2) but remain unsaturated or only saturate in some ensemble members for MSWX or CHIRPS driving data (Figure 4). The variation between different precipitation driving dataset ensemble medians is highest during the late pre-monsoon and early monsoon, reaching as much as 0.2 $m^3.m^{-3}$, but is much lower in the post-monsoon with all model outputs showing a similar soil moisture loss during drying. Importantly, the differences produced by different driving datasets are almost as large as the seasonal variation.

   To understand which parameter sets are optimal, we compare the individual ensemble members to the ground truth – the soil moisture record from our monitoring stations. First, we can evaluate the relative performance of each precipitation driving dataset. For the shallowest layer, expected to be the most sensitive to precipitation drivers, CHIRPS showed the best overall





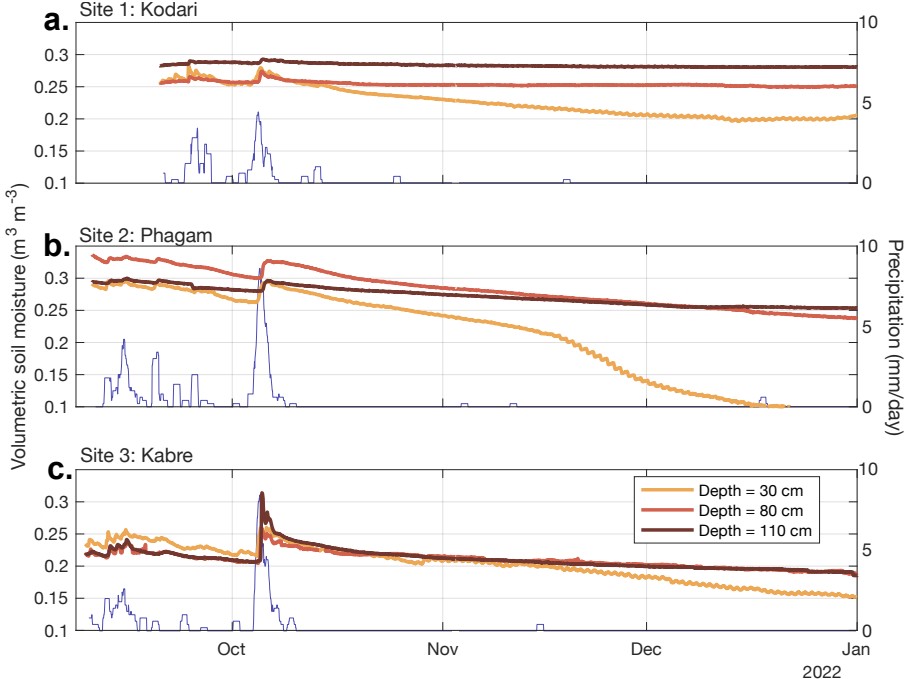

**Figure 3.** Soil moisture (orange lines) and precipitation (blue line) at the three field monitoring sites. Note the increase in soil moisture during large precipitation events. The shallowest soil level (30 cm) showed the most pronounced seasonal drying and a weak diurnal cycle.

performance across the three stations, with a median KGE of 0.13 (interquartile range IQR -0.24, 0.52 ; 95% confidence interval CI -0.99, 0.75) for Site 1, 0.35 (IQR 0.12, 0.46; 95% CI -0.25, 0.58) for Site 2, and 0.27 (IQR -0.15, 0.53; 95% CI -0.54, 0.74) for Site 3. Both the median and IQR are above the KGE mean field benchmark of -0.41. The median KGE of the MSWX ensemble is also above this threshold for all three stations, while GPM only meets this threshold for two of the three sites, and ERA5-land only for one. The better overall performance of CHIRPS may be expected due to its finer spatial resolution than the other datasets (Funk et al., 2015).

We create an 'optimal' parameter set by filtering each ensemble for the highest model-data KGE. Instead of selecting a single best ensemble member, we select all models with a KGE above the 95th percentile of the ensemble – a total of 50 individual members in our case. Selecting a subset of 50 members rather than a single one better accounts for the parametric uncertainty of both the model and comparison field data. This subset of 'best' models is shown as the blue line in Figure 5, relative to the overall ensemble in black, single run with default values in green, and field data in red. Note the relatively good fit between these best values and the field data, while a run without local calibration (using the default values) significantly overestimates soil moisture content. This soil moisture overestimation is consistently present regardless of the exact precipitation driving dataset chosen.

We also compare the model outputs to the field data by fitting an exponential decay to each, using equation 6. The fit between data and the exponential decay model for the quasi-rain-free period between 18 October and 18 December is excellent, with a



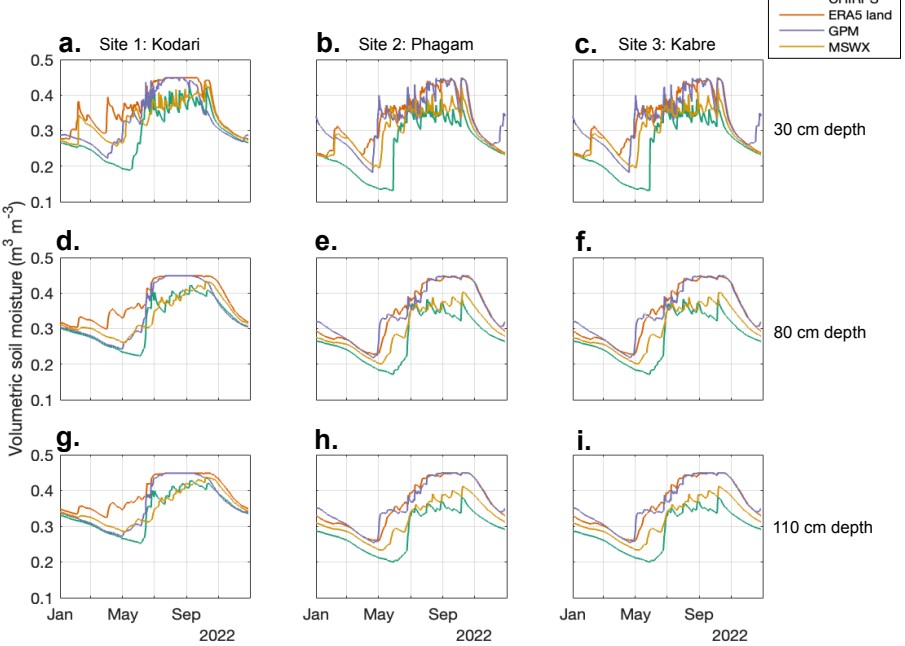

**Figure 4.** Median volumetric soil moisture for the Parameter Perturbation Experiment ensemble from the three different sites, the three different soil moisture depth levels investigated, and the four different precipitation driving datasets for all of 2022. The differences between the different soil moisture curves can be almost as large as the magnitude of the seasonal soil moisture cycle, highlighting the importance of selecting appropriate driving data.

coefficient of determination greater than 0.95 in all cases (greater than 0.99 for ensemble simulations driven by CHIRPS and MSWX) and a root mean squared error of less than 0.005 (Figure 6). The exponential decay term $\tau$ that controls the rate of
moisture drawdown has a broad range of values. The default parameter set results in a best fit $\tau$ of 688 day$^{-1}$ versus 1960 day$^{-1}$ for the field data, the latter representing a considerably slower drying of soils than the former. The majority of the ensemble members also predict more rapid drying than the field data, but a small number (14) of ensemble members also have a drying rate exponent $\tau$ which exceeds that of the field data.

### 3.3 Evaluation of parameter distributions

We may, therefore, use the results of this parameter perturbation experiment to determine an optimal model parameter set. We identify the best ensemble members either using the model-data KGE (Figure 5) or using the exponential decay cost function (Equation 7; Figure 6). We then back-calculate from each of the soil hydraulic parameters using each 'best fit' run. The full set of soil hydraulic parameters represents the prior distribution, while the two sets of 'best fit' parameters represent the posterior distributions as filtered by the Parameter Perturbation Experiment (Figure 7). The two posterior distributions, filtered by the
KGE and exponential decay function respectively, differ considerably from both the prior and default values but are generally



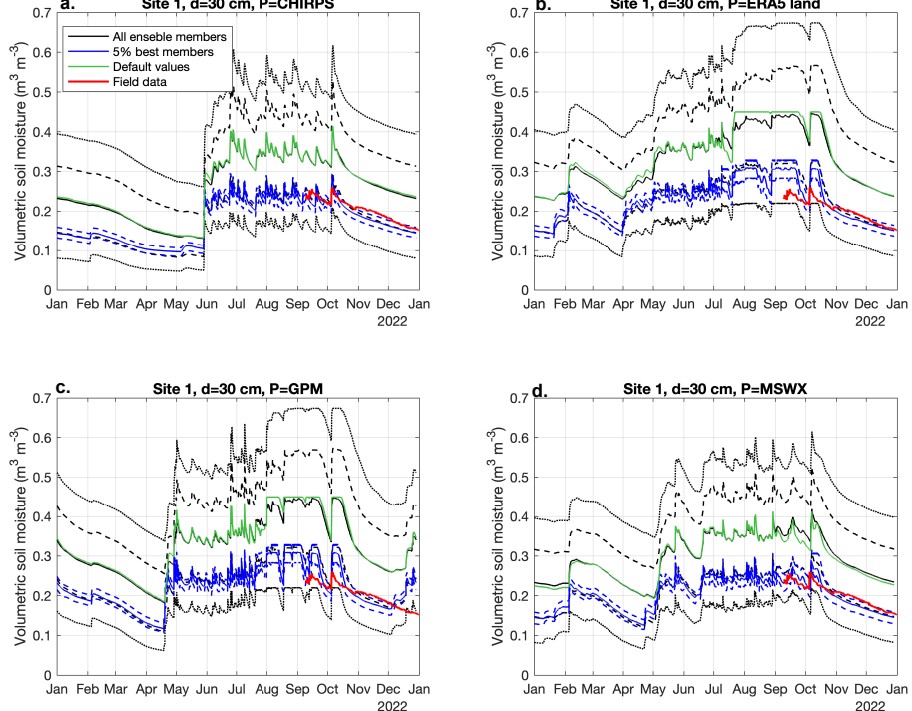

**Figure 5.** Comparison of the full ensemble spread and lowest Kling-Gupta Efficiency ensemble members with field measurements from 1 (Kodari). Note the varying degree to which the different precipitation datasets capture the early October rainfall event. Solid lines represent the median value, dashed lines represent the interquartile range, and dotted lines the 95% confidence interval.

similar, suggesting that both parameter selection methods are effective at improving the model representation at these locations. In particular, the posterior hydraulic conductivity ($K_S$) and thermal conductivity ($h_{con}$) are approximately a factor of 2 higher than the default, while the soil moisture at saturation ($\theta_S$) and critical soil moisture ($\theta_{crit}$) are approximately half of the default. Since the two methods produce broadly similar results, we propose that the KGE-based metric may be more effective as it still

applies outside the post-monsoon drawdown period, while the exponential decay fit may be ineffective in the event of a heavy rainfall event causing significant soil moisture recharge.

## 3.4 Evaluation across stations

While the determination of parameter sets with a close match to the field data at a given field site demonstrates that the model can effectively represent soil moisture changes, it does not provide any information about whether this parameterization can

be applied to a broader region. To do this, we must evaluate an optimal parameter set from one site with the field data from another site. We use posterior parameter sets from each field site to run a 50-member ensemble at the other two field sites and determined whether the resulting soil moisture curves fit the field data better than the model default (Figure 8). For example,




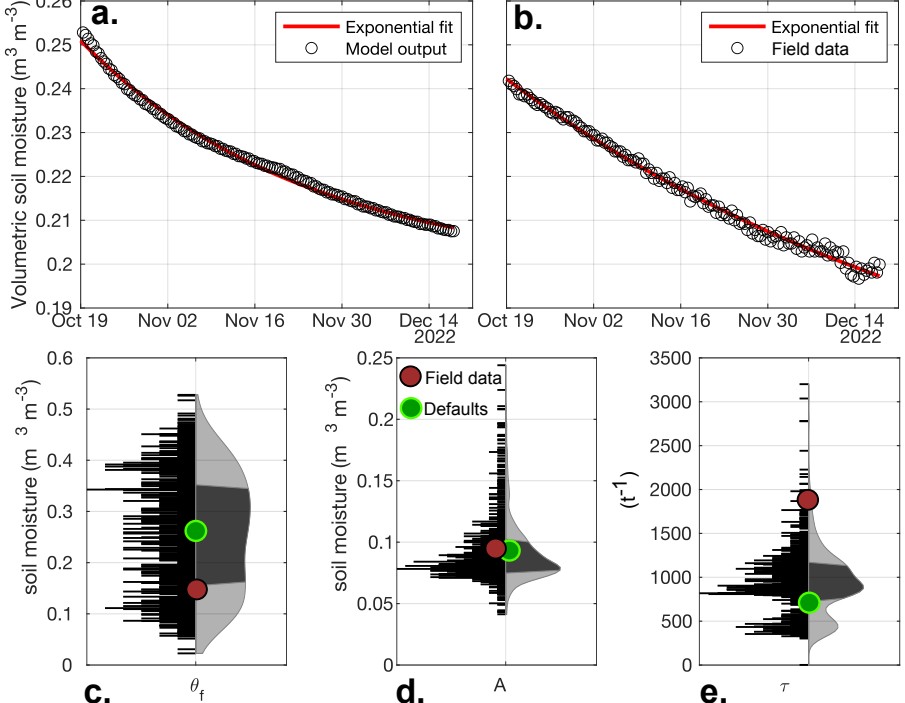

**Figure 6.** Exponential fit to the post-monsoon dry down from 18 October to 18 December 2022 for the model and field data at Site 1 (Kodari) for CHIRPS driving data and the shallowest layer. The exponential fits to this portion of the data are excellent, with a coefficient of determination of 0.95 or greater in 99.9% of ensemble members and the field data. The three lower subplots show the three fitting parameters of the exponential model (e.g., Shellito et al., 2016), with the left side of the violin plot showing the full data histogram and the right side showing the distribution with interquartile range (dark grey) and 95% confidence intervals (light grey). The red and green dots show the field data and default value.

we take the best parameter sets calibrated from Site 3 and apply these to Site 1, comparing the resulting distribution to the prior and default. In this experiment, the updated ensemble provides an independent test of the effectiveness of the model calibration.

The model parameterization from Site 1 (Kodari) provides an improvement at both of the other field sites at all three soil moisture depth levels, with all KGE values above the threshold of -0.41 (Knoben et al., 2019) except the 30 cm soil moisture layer at Site 2. KGE values are extremely low (-3 to -15) for the two lower soil moisture levels for Site 1 (Kodari) both with the full ensemble spread and parameterized with the best fit from the other two sites. The low KGE values are primarily related to the extremely low variance of instrumental soil moisture values for these sites (0.0249 and 0.0125 $m^3.m^{-3}$ for 80 cm depth and

110 cm depth respectively) compared to field data (0.204 and 0.171 $m^3.m^{-3}$ for 80 cm depth and 110 cm depth respectively). Both parameterizations from Site 2 (Phagam) and 3 (Kabre) improve the model-data fit (higher KGE) for the 30 cm depth soil layer. The improvement is greatest for Site 3 (Kabre), just as the improvement at Site 3 is the best with the parameterization from Site 1.



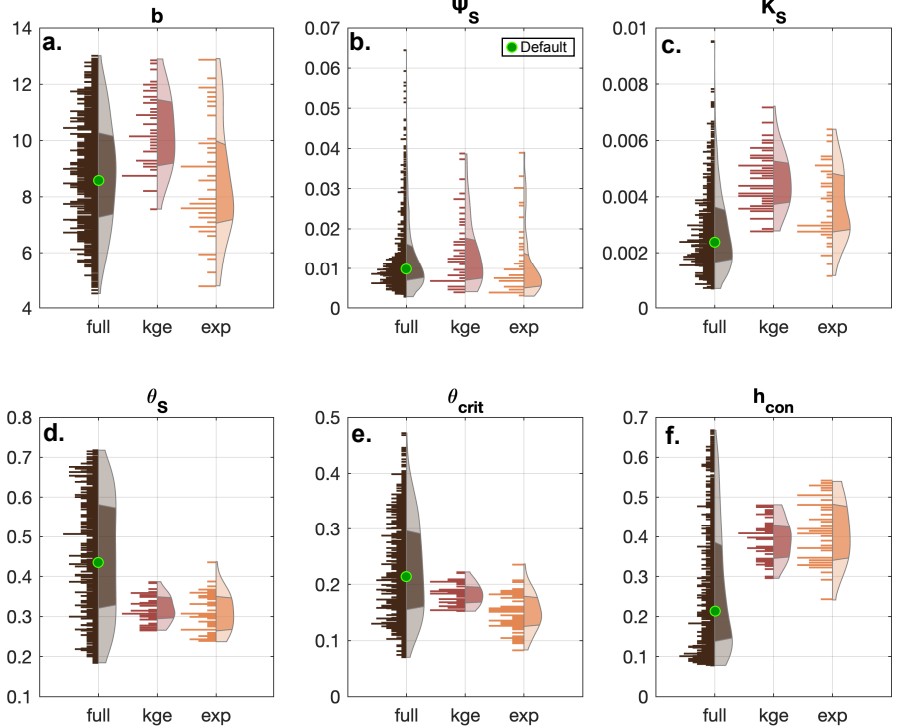

**Figure 7.** Prior ('full') and posterior ('kge' and 'exp') soil hydraulic parameter distributions resulting from the parameter perturbation experiment for Site 1 (Kodari). The 'kge' posterior was constructed from the 95th percentile of the Kling-Gupta Efficiency values, while the 'exp' posterior was constructed from the 95th percentile of the exponential fit cost function.

As well as using the KGE, we may evaluate the improvement in the model fit by calculating the root mean squared error (RMSE). The RMSE for the full ensemble at 30 cm depth at Site 1 is 0.1466 $m^3.m^{-3}$, compared to 0.0683 $m^3.m^{-3}$ for the best parameter set from Site 2 and 0.0431 $m^3.m^{-3}$ for the best parameter set from Site 3. Similarly, the RMSE for the full ensemble at 30 cm depth at Site 3 is 0.1354 $m^3.m^{-3}$, compared to 0.0436 $m^3.m^{-3}$ for the best parameter set from Site 1 and 0.0635 $m^3.m^{-3}$ for the best parameter set from Site 2. Overall, we find that JULES provides a substantially more accurate soil moisture record when calibrated at the other field sites relative to default values.

## 4 Discussion

Both our instrumental and modelled results show a continuous drying of soils at our three field locations post monsoon. For each site, we run a 1000-member ensemble varying soil pedotransfer constants and associated hydraulic parameters and using four different precipitation driving datasets. We evaluate the fit between this range of different model soil moisture trajectories and field data using the KGE and an exponential fit to both. Finally, we use cross-comparisons between the best parameter values for the three stations to evaluate the transferability of a local optimised parameter set to a regional or national-scale





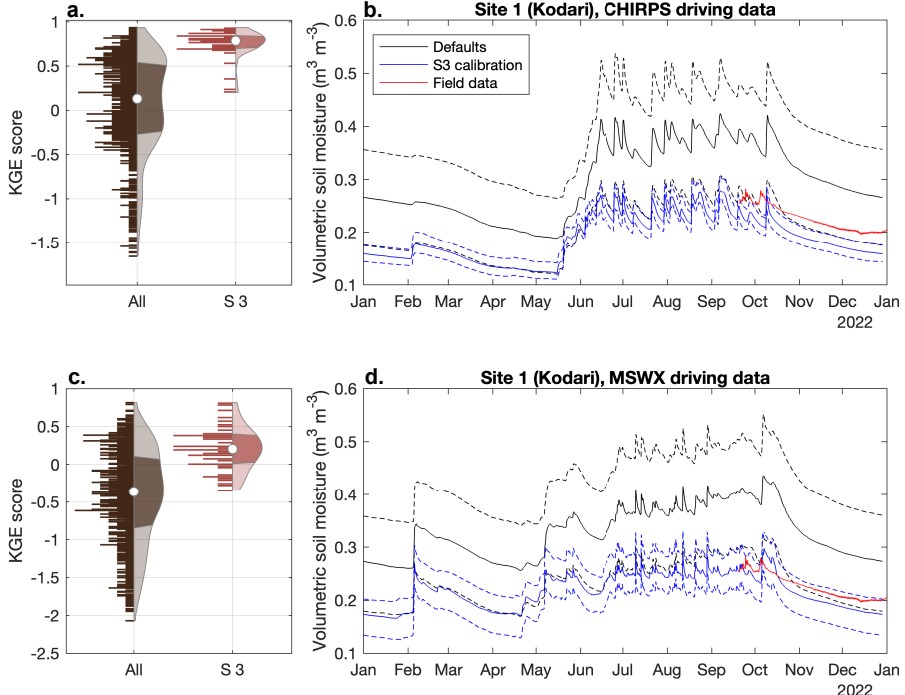

**Figure 8.** Comparison between the full ensemble run for Site 1 ("All") and the best ensemble member parameter set derived from Site 3 (Kabre). The ensemble members shown in this plot were driven using CHIRPS precipitation data.

study. In all cases, except those with known complicating factors, we find that the parameterization from a different station more closely matches the field data than the default model parameters, suggesting that the parameterizations are applicable over a broader region.

Our two calibration methods, the direct time series comparison using the KGE and the comparison of exponential fit param-
eters, both yield similar results. While both provide slightly different information, the direct time series comparison is more versatile as the exponential decay comparison cannot be used outside of the period of post-monsoon soil moisture loss, and will likely fail in the event of a rare late (October-December) rainstorm. Since our calibration workflow method uses point runs of the JULES land surface model, it remains computationally efficient even with a large number of ensemble members ($10^3$) capturing the full parameter space. Additionally, this method can easily integrate additional information from new field
monitoring stations where they become available either for testing an existing calibrated model in a new location or for further refining the best model calibration. Finally, since this method directly calibrates pedotransfer function constants instead of directly fitting soil hydraulic parameters, it can easily integrate information about spatially varying soil physical properties.

Our use of four different precipitation driving datasets enables us to evaluate the uncertainties inherent in reanalysis and remotely-sensed datasets over complex mountainous terrain, and the relative advantages and disadvantages of CHIRPS, ERA5-
land, GPM-IMERG, and MSWX in terms of anticipating the evolution of soil moisture. Our first finding is that there are large





variations in both the magnitude and seasonality of the resulting soil moisture curves depending on the precipitation dataset used (e.g., Figure 4), with differences between datasets almost as large as the magnitude of the seasonal variation within any single dataset. We, therefore, recommend that any study either use multiple precipitation driving datasets as we do here or carefully compare any single precipitation dataset to a range of field-based precipitation measurements to correct for biases in

the data. Overall, CHIRPS and MSWX provide a better fit to the data than ERA5-land and GPM-IMERG, although this is not the case across all ensemble members. CHIRPS has a higher spatial resolution (0.05 degrees; 5 km) relative to the other three datasets (0.1 degrees; 10 km) and may better capture the strong spatial gradients in the complex Nepali topography (Funk et al., 2015).

Our method has several potential limitations, both from uncertainties in the datasets used and from our chosen modelling

and parameter estimation framework. The first limitation is related to uncertainties and possible instrument errors in the field data used for the selection of the best parameterization. Any errors related to instrumental drift, poor contact between the probe and soil, or other uncertainties will lead to biases being transferred into the model parameter calibration. We mitigate this bias with information on sensor status from the field (e.g., battery voltage) and detailed manual inspection of field data for potential anomalies. The second potential issue, as discussed in the previous paragraph, is related to any biases or errors in the datasets

used to drive the land-surface model. For instance, model driving data with higher precipitation values than reality may result in incorrect optimal soil hydraulic parameters to compensate for this bias. We address this issue by using a suite of four different possible precipitation datasets and recommend taking the same approach for any further regional-scale calculations used to determine optimal model parameters to minimise potential bias.

Limitations related to our parameter estimation and modelling framework are introduced at different stages of the modelling:

the choice of pedotransfer function, the choice of parameter determination method, and physical processes not represented in JULES. A range of different pedotransfer functions can be used to map soil physical characteristics to hydraulic parameters, with best-suited functions depending on data availability and modelling requirements (e.g., Tomasella and Hodnett, 2004; Van Looy et al., 2017). We use the Cosby pedotransfer functions, which were empirically constructed based on a US soil database but have been widely applied due to their simplicity and flexibility (Cosby et al., 1984). The soil texture data which we

use, derived from SoilGrids 2.0 (Poggio et al., 2021; de Sousa et al., 2020), may also be uncertain due to the relatively sparse field validation in Nepal, which can in turn bias the pedotransfer function constants in compensation. We use a perturbed parameter ensemble approach to calibrate JULES based on a Latin hypercube sampling. This method is computationally efficient and largely insensitive to local minima, but cannot account for possible parameter values outside of the range sampled in the Latin hypercube (here $\pm$ 50%). Finally, the JULES land surface model itself provides only a partial representation of the

full suite of hydrological processes affecting the infiltration and movement of water through the soil. For a full discussion of the processes currently included in JULES, we refer readers to the model documentation (e.g., Best et al., 2011; Clark et al., 2011). Groundwater flow and lateral water flow within the soil layer is one process that is not accounted for and may be locally important.

Once an optimal parameterization has been established, this may be used to model soil moisture changes in an area with

no available field data. In our workflow, we select a set of 50 (95th percentile out of 1000 members) optimal parameters from





our ensemble. We explore an example of this here and model our best estimate of volumetric soil moisture at Laprak, Nepal. Laprak is a village in Gorkha district of central Nepal located approximately halfway between Sites 1 (Kodari) and Sites 2 and 3 (Phagam and Kabre; see Figure 1 for location). The town of Laprak, built on top of a slow-moving landslide (GURUNG et al., 2011; Haneberg et al., 2022; Khanal, 2007), was heavily damaged during the 2015 Gorkha earthquake and has since

been relocated to a new, higher elevation site. Understanding soil moisture content at Laprak can therefore help establish the baseline risk of shallow landsliding, and possibly also provide insight into the likelihood or rate of deformation of the slow-moving landslide (Lacroix et al., 2020). We model soil moisture at Laprak from January to December 2022 using the 50 optimal ensemble members for dach of the two precipitation driving datasets: CHIRPS and MSWX (Figure 9). We also model soil moisture change using the default parameter set using both precipitation datasets – this can establish the baseline

that would be calculated using JULES in the absence of local calibration. The median volumetric soil moisture of the resulting 100-member ensemble (combining the two precipitation driving datasets) is lower than both default parameter runs, in some cases by as much as 50% (Figure 9). This provides an uncertainty-bounded time series that can be used for a range of follow-up applications, including comparison with independent records of hillslope stability or landslide occurrence to establish the nature and extent of correlation.

Landslides are not independent of other hazards and are commonly triggered by and can themselves initiate or compound other hazards as part of a multi-hazard chain (Aksha et al., 2020; Bell and Glade, 2012; Kappes et al., 2012; Zhang and Zhang, 2017). For example, earthquakes may trigger landslides (e.g., Chigira et al., 2010; Kincey et al., 2021), and landslide deposits may transition into highly-mobile debris flows or dam and flood river networks (Fan et al., 2020; Shugar et al., 2021). Comparative studies have shown that the degree of interaction between hazards may vary greatly between sites, for

instance with the similar magnitude 2008 Wenchuan earthquake in China triggering approximately an order of magnitude more landslides than the 2015 Gorkha earthquake (Xu et al., 2016). Preconditioning factors such as soil moisture can play an important role in these interactions: saturated soil will be more likely to fail at a given seismic peak-ground-acceleration than dry soil, and any resulting landslide is expected to have a greater mobility and runout distance (Bogaard and Greco, 2016; Keefer, 1999; Marc et al., 2016; Záruba and Mencl, 2014). For example, an earthquake occurring during or immediately

following Nepal's monsoon might be expected to trigger a larger number of landslides than one occurring before it, as with the 2015 Gorkha event. Systematic and large-scale mapping of soil moisture in landslide-prone areas can help improve our understanding of baseline conditions for multihazard events, and improve our hazard and risk estimates for other hazards such as earthquakes and debris flows.

Building on this work, future studies of soil moisture and landsliding may focus on some of the following outstanding issues:

(i) Increasing the spatial (number of stations) and temporal (duration of each record) availability of field measurements of soil moisture for model evaluation and parameter determination; (ii) Upscaling of soil moisture measurements to a regional or national scale using derived parameterizations; (iii) Operationalizing the use of soil moisture estimates in landslide forecasting, for instance, through the use of global weather forecast data as driving data for JULES. Our use of the JULES, which is already integrated into the Met Office Unified Model framework, reduces the barriers to operationalizing soil moisture calculations



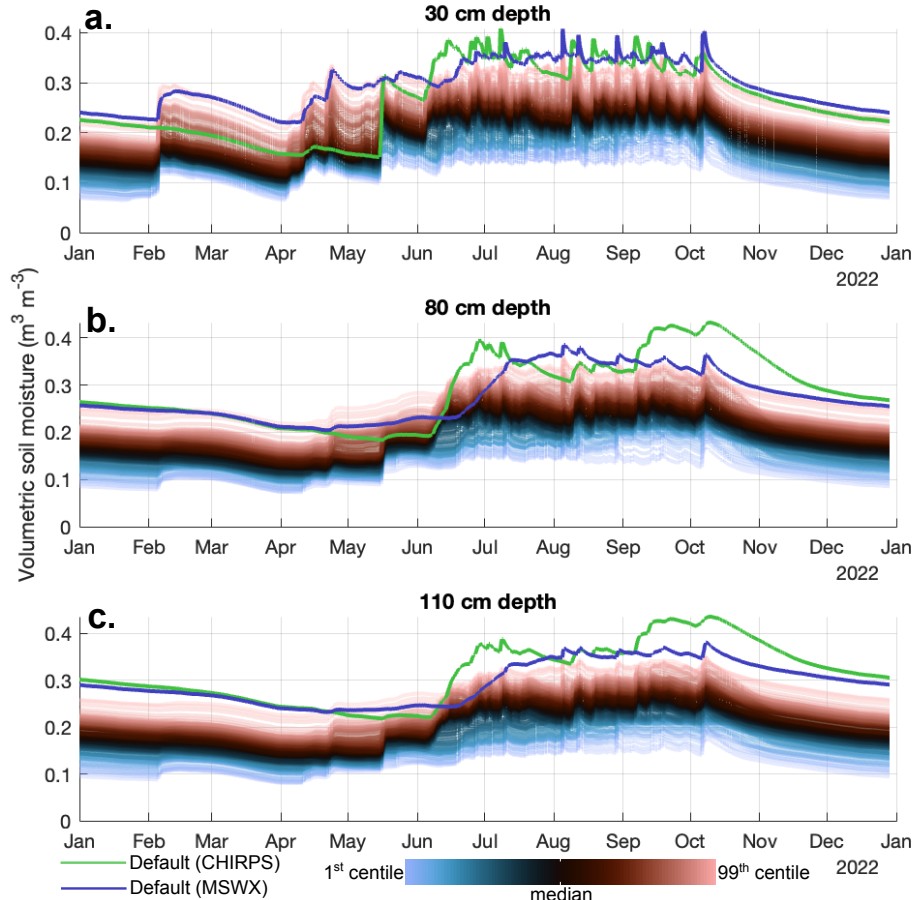

**Figure 9.** Multi-source ensemble soil moisture for an unmonitored site, Laprak, using the best parameter sets from Site 1 for MSWX and CHIRPS precipitation driving data. The solid lines show the runs with default parameters, while the colour field shows the ensemble run with calibrated best-fit parameters. The ensemble-based approach provides both the overall trend and a measure of the uncertainty in possible soil moisture values.

with real-time forecast data; and (iv) Hindcasting of soil moisture conditions at known historical landslides for improving our understanding of the soil moisture conditions conducive to hillslope failure across a wide range of geographic environments.

## 5    Conclusions

Soil moisture is an important physical parameter for evaluating spatial and temporal changes in hillslope stability and pre-disposing factors for landsliding. Here, we develop a modelling framework for parameterising the JULES land surface model

using volumetric soil moisture time series from three field monitoring stations. All three field sites are in Nepal, with one in Sindhupalchok district (Kodari, 50 km NE of Kathmandu) and two in Myagdi district (Phagam and Kabre, 200 km NW of



Kathmandu). We use Latin hypercube sampling of pedotransfer function constants to calibrate soil hydraulic parameters and run a 1000-member ensemble for each site and all four different precipitation driving datasets. All model members reproduce the distinct soil moisture seasonality in Nepal, and we use two different metrics, the KGE and exponential fit constants, to

identify the models with the best fit to our field data. We then use the best-fit parameters from each site to model soil moisture change at the other two sites, with the calibrated model having a 50% or greater reduction in RMSE relative to the default parameter sets. We interpret this improved fit between spatially distant locations as evidence that the parametrisations are applicable on a regional scale, and can be used in areas for which field data is not available. Better soil moisture nowcasts and forecasts, as enabled by the improved parameter estimation showcased in this study, can improve our understanding of transient

landslide susceptibility and the interactions between landslides and other hazards.

*Author contributions.* MV and SD conceived the project based on discussions with the entire team. MV conducted the modelling and drafted the initial manuscript. All authors commented on the analyses and final manuscript. The sensor network this manuscript is based upon has been maintained by many of the author team.

*Competing interests.* The authors declare no competing interests.

*Acknowledgements.* This research was supported by a grant from the UKRI Global Challenges Research Fund (NE/T01038X/1). We acknowledge discussions with colleagues at the UK Centre for Ecology and Hydrology (CEH) about soil moisture calibration. We thank the broader Sajag Nepal project team for various discussions around the instrumentation and hazard implications of this work.



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
