# Peer review of "Modelling of post-monsoon drying in Nepal: implications for landslide hazard"

_EGUsphere, 2024_

## Referee Comment (RC1)

Review of the manuscript Modelling of post-monsoon drying in Nepal: implications for landslide hazard by Maximillian Van Wyk de Vries et al.

**Summary**

The Authors present and discuss a study about soil moisture monitoring and modelling. The analysis is performed based on a dataset of point-scale soil moisture observations collected at three locations and at three depths in Nepal. The land-surface model JULES has been calibrated based on two different criteria. The Authors further discussed the results in the light of landslide predictions.

**General comment**

I acknowledge that the Authors made a good effort to shape a scientific manuscript out of the collected data-set and modelling exercises. Despite this effort, however, I regret to say that I personally believe that the scientific value is weak in many aspects as detailed below. While trying to be constructive with my critiques, I feel that a completely different manuscript should be prepared, and the current manuscript should not be foreseen.

**Specific comments**

[1] Landslide topic not addressed.

The Authors promote the landslide prediction as motivation of the study. While I agree that soil moisture is an important trigger in many conditions, by reading this manuscript I believe that this is not the most relevant factor for landslide prediction in this specific environment. Moreover, at the end, the landslide topic is not addressed at all. Specifically, we are in a monsoon area with wet and dry period. So, discussion about when landslides mainly occurred should be reported. The Authors show how precipitation based on different products strongly varies and affect soil moisture prediction (L311). So, I'm expecting that getting precipitation right is more relevant and calibrating the model to the observed soil moisture with wrong forces could be misleading, i.e., compensating the error by calibration (L326). Finally, how much the use of observations and of the calibrated model improves landslides prediction is not at all implemented. Strictly speaking, why not use a landslide/erosion model instead of JULES if the objective is to cover the landslide topic?

[2] Soil moisture network weak

Despite I believe that installation and maintenance of soil moisture sensors at these sites is challenging, it should be acknowledged that the data-set is quite limited. Without pretending the installation of other sensors, the Authors seems to have data from sep-22 till now. Surprisingly they use and show only data of 2022. Why? The analysis should be at least extended to the entire 2023 having then two drying seasons. Moreover, point-scale sensors

are used, and no discussion is reported about their representativeness. It could be likely the case that installing the sensors one meter apart could show a completely different behavior. Pretending one profile of point-scale sensor to be a ground truth when driving forces are at 5-10 km resolution is questionable. Many studies working on spatial mismatches have been conducted and should be considered for better shaping the study and extending the discussion.

[3] modelling exercise

The model and the modelling framework is not new, to some extent confuse and it does not provide any new insights on the use and capability of these models, especially for supporting landslides predictions. More specifically, a spatial mismatch between point-scale sensors and modelling is critical and is not addressed. The use of JULES for addressing landslide prediction is misleading. Why using this model? It is also not clear to me why the Authors need the exponential decay function. Could you not directly calibrate the model parameters by looking at the dry down period? I do not expect to see different results than first fitting the decay function. Why even testing that if at the end the Authors argue that it is not a good approach to follow (L301)? The use of different precipitation sources seems to disappear at a certain point, i.e., 3.4 evaluation of parameter distribution is discussed but is not clear which precipitation product is used. Comparison between the distributions obtained based on different precipitation products could shed light on the importance of the driving factors more than soil moisture. The use of measured precipitation should also be foreseen. Evaluation across stations is a good exercise. I would strongly suggest also testing during another period, i.e., 2023? Discussion across precipitation products is missing. RMSE suddenly appears at L285. The exercise shown at figure 9 is not clear. Which best parameter sets is used? From site 1, 2 or 3? All? Why selecting two precipitation products? Overall, the modelling approach fails to quantify if the uncertainty in driving data is more relevant than uncertainty in model parameters and if a model can better predict landslides if soil moisture observations are used.

[4] Clarity weak

The manuscript is in general well written, but I found some passages difficult to understand and many parts where description should be improved. E.g.,

The text at L14-24 is difficult to grasp, it focuses on Nepal and some few general statements anticipating the objectives of the manuscript. The more general introduction seems to start at L25 where the general topic is introduced. The above text might be better integrated later.

L73. I do not think there is a clear definition of how many sensors make a network, but I was expecting more than 3 locations with 3 depths for a network.

Figure 2. Description in the legend caption should point to (a) (b) etc. I do not understand why cumulative precipitations are plotted but the values decrease. If it is a cumulative precipitation I should see monotonic increase. These would also better capture the difference between the precipitation products. What is the meaning here to plot field if they start from September? Why not also using 2023?

L125. How is the soil discretized in the model? Do we have three soil layers according to the soil moisture sensor depth?

L178. If no precipitation occurs during post-monsoon, is there any landslide risk over this period? It is not clear from the manuscript why modelling this behavior is important underpinning the scientific value of this study.

L195. Assessing the value of this study by looking at how much the prediction of soil moisture dynamics increases from the uncalibrated model is misleading. If the objective is to understand landslide predictions, the Authors should show how landslide prediction improves by improving soil moisture modelling.

Figure 4 shows, as far as I understand, only modelled data and high spread of the modelled soil moisture due the different precipitations products. This could support the conclusions that uncertainty in precipitation might be more relevant than soil moisture for landslide predictions.

Figure 5 has too many lines, and it is difficult to read, in my opinion.

Figure 6 is not discussed in the main text. Only cited at L257. Is it then useful? What is default value here?

L275. As far as I have understood the Authors only show the use of the best set of parameters from Site 3 and apply to site 1 (Figure 8). Results over the other combinations are not shown but are relevant. In figure 8 I would have also expected to see a comparison to the KGE distribution obtained based on the best ensemble member from site 1. Legend caption of Figure 8 says that ensemble members were driven using CHIRPS but the plot d says also MSWX driving data.

---

## Author Comment (AC2)

**Modelling of post-monsoon drying in Nepal: implications for landslide hazard**

Maximillian Van Wyk de Vries[1,2,3], Sihan Li[4], Katherine Arrell[5], Jeevan Baniya[6], Dipak Basnet[6], Gopi K. Basyal[7], Nyima Dorjee Bhotia[6], Alexander L. Densmore[8], Tek Bahadur Dong[6], Alexandre Dunant[8], Erin L. Harvey[8], Ganesh K. Jimee[7], Mark E. Kincey[9], Katie Oven[5], Sarmila Paudyal[7], Dammar Singh Pujara[7], Anuradha Puri[6], Ram Shrestha[7], Nick J. Rosser[8], and Simon J. Dadson[3]

[1]Department of Geography, University of Cambridge, Cambridge CB2 3EL, UK.
[2]Department of Earth Sciences, University of Cambridge, Cambridge CB3 0EZ, UK.
[3]School of Geography and the Environment, University of Oxford, Oxford OX1 3QY, UK.
[4]Department of Geography, University of Sheffield, Winter St, Sheffield S3 7ND, UK.
[5]Geography and Environmental Sciences, Northumbria University, City Campus, Newcastle upon Tyne, NE1 7RU, UK.
[6]Social Science Baha, 345 Ramchandra Marg, Battisputali, Kathmandu, Nepal.
[7]National Society for Earthquake Technology, Sainbu Bhainsepati Residential Area, Lalitpur 13775, Nepal.
[8]Department of Geography, Durham University, Lower Mountjoy, South Rd, Durham DH1 3LE, UK.
[9]School of Geography, Politic, and Sociology, Newcastle University, Newcastle upon Tyne NE1 7RX, UK.

Correspondence: M. Van Wyk de Vries (msv27@cam.ac.uk)

**Response to Reviews**

We thank both reviewers for their comments and respond between the lines in red below.

Review 1

Dear Reviewer,

We sincerely thank you for your thorough and constructive review of our manuscript. Your insights have been invaluable in improving the clarity and impact of our work. We have carefully considered all your comments and have revised the manuscript accordingly. Below, we provide detailed responses to each of your specific comments.

Review of the manuscript
Modelling of post-monsoon drying in Nepal: implications for landslide hazard
by Maximillian Van Wyk de Vries et al.

Summary
The Authors present and discuss a study about soil moisture monitoring and modelling. The analysis is performed based on a dataset of point-scale soil moisture observations collected at three locations and at three depths in Nepal. The land-surface model JULES has been calibrated based on two different criteria. The Authors further discuss the results in the light of landslide predictions.

General comment

I acknowledge that the Authors made a good effort to shape a scientific manuscript out of the collected dataset and modelling exercises. Despite this effort, however, I regret to say that I personally believe the scientific value is weak in many aspects, as detailed below. While I am trying to be constructive with my critiques, I feel that a completely different manuscript should be prepared, and the current manuscript should not be foreseen.

We appreciate your candid feedback. We have taken your comments seriously and have substantially revised the manuscript to strengthen its scientific value.

In particular, we feel that the key message from this manuscript, and the reason for sending this to journal SOIL instead of a landslide/hazard journal, was not entirely clear. It is a well-established fact from geotechnical engineering and soil mechanics that soil moisture contributes to landslide triggering. However, in the present day, a very small proportion of landslide monitoring or forecasting tools use this information. The main reason for this is that observations are generally absent, unreliable, or on a scale inapplicable to landslide monitoring. Before landslide and soil moisture data can be directly compared, there is a need to identify methodologies to compile this data on a suitable spatial scale - and with sufficient accuracy.

This study does this - in one of the regions of the world that has both the fewest direct soil moisture measurements and the most landslides. Acknowledging the limitations of the datasets used, we consider that this manuscript provides important confirmation of the potential form upscaling from point measurements using calibrated land-surface models. In particular, our results are useful for understanding the "post-monsoon decay", associated with increasingly deadly landslide episodes in recent years, including in late September and early October this year (2024).

We hope that the revised manuscript addresses your concerns and demonstrates the significance of our study and its relevance to this journal.

Specific comments
Landslide topic not addressed
The Authors promote landslide prediction as the motivation for the study. While I agree that soil moisture is an important trigger in many conditions, by reading this manuscript, I believe that this is not the most relevant factor for landslide prediction in this specific environment. Moreover, at the end, the landslide topic is not addressed at all. Specifically, we are in a monsoon area with wet and dry periods, so a discussion about when landslides mainly occur should be reported. The Authors show how precipitation, based on different products, strongly varies and affects soil moisture prediction (L311). Therefore, I expect that getting precipitation right is more relevant, and calibrating the model to the observed soil moisture with incorrect forces could be misleading, i.e., compensating the error by calibration (L326). Finally, the use of

observations and the calibrated model to improve landslide prediction is not at all implemented. Strictly speaking, why not use a landslide/erosion model instead of JULES if the objective is to cover the landslide topic?

We thank the reviewer for this comment and have sought to expand and clarify our discussion around landslides. Indeed, as we discuss here, we consider this work to be an important component of future work on landslide susceptibility in the post-monsoon period of Nepal, even though we do not directly utilise landslide data.

We agree that precipitation is a key factor when forecasting landslides. It is also well established that soil moisture is an important preconditioning factor for the impact of this precipitation - this understanding is, for instance, parametrised by intensity-duration curves. We do not claim that soil moisture is more important than precipitation here (on the contrary), this work instead aims to improve our understanding of soil mosture that may modulate the impact of precipitation (and other landslide triggers, such as earthquakes).

We hope that the clarifications to the objective of this manuscript clear up this point.

Soil moisture network weak
Despite the challenges of installing and maintaining soil moisture sensors at these sites, it should be acknowledged that the dataset is quite limited. Without expecting the installation of additional sensors, the Authors seem to have data from September 2022 till now. Surprisingly, they use and show only data from 2022. Why? The analysis should be extended to include the entire 2023 dataset, covering two drying seasons. Moreover, point-scale sensors are used, and no discussion is provided about their representativeness. Installing the sensors one meter apart could potentially show completely different behavior. Pretending one profile of point-scale sensors represents ground truth when driving forces are at a 5-10 km resolution is questionable. Many studies on spatial mismatches have been conducted and should be considered to improve the study and extend the discussion.

We thank the reviewer for this comment about the data used in this study. Indeed, the data was drawn from a theoretically larger network of 8 stations installed in two transects across the country, but consistent data was only available from three of these. This reflects an inherent challenge of collecting soil moisture data in remote, high-mountain environments. Indeed, larger or more complete datasets would always be preferable, and the key question is perhaps whether the data is sufficient to answer the questions we pose. In this case, the answer is yes, and the three stations provide us sufficient spread to test the land-surface-model based extrapolation into new areas using representative parameter sets. We can show that the parameterisations are applicable on a broader regional scale, greater than the resolution of the driving data (e.g. the 5-10 km resolution of precipitation).

Modelling exercise
The model and modelling framework are not new, somewhat confusing, and do not provide new insights into the use and capability of these models, especially for supporting landslide predictions. Specifically, the spatial mismatch between point-scale sensors and modelling is critical and is not addressed. The use of JULES for addressing landslide prediction is misleading. Why use this model?

We have adjusted our description of the study as this comment suggests the key reasons for choosing JULES were not clear. Indeed, we do not seek to provide new insight into the model, on the contrary, we select JULES as it has already been extensively tested. JULES provides a robust, flexible, and scalable model for large-scale soil moisture computation, and has the potential to be directly integrated with numerical weather prediction models. We have added additional information to the manuscript about the reasons for this choice.

It is also unclear why the Authors need the exponential decay function. Could they not directly calibrate the model parameters by observing the dry-down period? I do not expect different results compared to first fitting the decay function. Why even test this approach if, at the end, the Authors argue that it is not a good approach to follow (L301)?

The exponential decay represented a reduced-complexity approach to fitting the model outputs. If it were successful, it might have enabled simpler or more computationally efficient processing. We therefore maintain this portion of the manuscript as a test of this hypothesis, despite the KGE fit being more effective.

The use of different precipitation sources seems to disappear at a certain point, i.e., in section 3.4, where the evaluation of parameter distribution is discussed but without clarity on which precipitation product is used. Comparison of distributions based on different precipitation products could shed more light on the importance of driving factors than soil moisture alone. The use of measured precipitation should also be considered. While evaluation across stations is a good exercise, I would strongly suggest also testing during another period, such as 2023. Discussion of precipitation products is missing. RMSE suddenly appears at L285. The exercise shown in Figure 9 is unclear. Which best parameter set is used? From site 1, 2, or 3? All? Why select only two precipitation products? Overall, the modelling approach fails to quantify if uncertainty in driving data is more relevant than uncertainty in model parameters and if a model can better predict landslides with soil moisture observations.

We have added additional text to the manuscript to clarify each of these points, in particular:

-Clarify which precipitation dataset is shown in the plots in each case, and discussing more extensively the different datasets used.
-The exercise shown in Figure 9, which shows the estimation of possible soil moisture at a location with no local field data. The comparison exercise shows that calibrated models differ substantially from uncalibrated ones, and that this approach can produce
-Explaining why RMSE was an appropriate metric to use.

We agree that there are multiple sources of uncertainty sourced from both the driving data and model parameterisation - here we aim to isolate the parametrisation-related error by comparing across sites using the same driving data. We consider uncertainty in driving data to be external to the analyses conducted here but agree that understanding this is critical to both the soil-moisture and precipitation related elements of landslide susceptibility.

Clarity weak
The manuscript is generally well-written, but some passages are difficult to understand, and many parts require improved description. For example

We thank the reviewer for these detailed line by line comments and have made a number of changes to improve the overall clarity of the manuscript.

The text in L14-24 is difficult to grasp, focusing on Nepal with some general statements anticipating the manuscript's objectives. The general introduction seems to start at L25, where the broader topic is introduced. The earlier text might be better integrated later.

We have reorganized the introduction to provide a clearer progression from the general context of landslide hazards in Nepal to the specific objectives of our study. The initial paragraphs now establish the significance of soil moisture in landslide susceptibility before introducing our modeling approach.

L73: While I do not think there is a clear definition of how many sensors constitute a network, I expected more than 3 locations with 3 depths for a network.
We consider that the term "network" is appropriate here, and that the text and figure (1) clearly show the number and spatial distribution of these stations.

Figure 2: The description in the legend caption should point to (a), (b), etc. I do not understand why cumulative precipitations are plotted with decreasing values. If it is cumulative precipitation, there should be a monotonic increase, which would better capture the differences between the precipitation products. Also, why only plot data starting from September? Why not use data from 2023?

The y axis label was accidentally mislabeled and has now been changed to clarify. The data is not cumulative but is instead continuous with a monthly running mean. As mentioned in the figure caption, the comparative field data was not available outside of this time window, thus the bounds of this plot.

L125: How is the soil discretized in the model? Do we have three soil layers corresponding to the soil moisture sensor depths?
Yes, we apply a transformation to the Jules outputs to obtain data at the sensor depth. This is described l175 ("We linearly interpolate between the JULES model layer depths (5±5 cm, 22.5±12.5 cm, 67.5±32.5 cm, and 200±100 cm) to obtain soil moisture at the exact depths of the field measurements (30, 80, and 110 cm; Figure 1)").

L178: If no precipitation occurs during the post-monsoon period, is there any landslide risk? It is unclear why modelling this behavior is important for the scientific value of this study.
Landslide triggers are multifactorial, including heavy precipitation, seismicity (for instance, the 2015 Gorka earthquake was devastating in this region), and anthropogenic activity. Soil moisture contributes to any of these.

Furthermore, the assumption here would still be valid if "virtually no precipitation occurs" before a heavy precipitation event triggering landslides. The exponential decay would break down after this but remain valid in the preceding time and be useful for understanding soil moisture preconditioning.

L195: Assessing the value of this study by looking at the improvement in soil moisture dynamics from the uncalibrated model is misleading. If the objective is to understand landslide predictions, the Authors should show how landslide prediction improves with improved soil moisture modelling.
As mentioned above, we aim to clarify the objective of this study. Directly showing a link between landslide prediction and the soil moisture parametrisation is beyond the scope of this work. However, showing an improvement in our ability to model soil moisture dynamics is an important step towards this, particularly in data-starved regions.

Figure 4 shows only modelled data, as I understand, with a high spread of modelled soil moisture due to different precipitation products. This could support the conclusion that uncertainty in precipitation is more relevant than soil moisture for landslide predictions.

We thank the reviewer for this comment which we agree with in part. Variation between precipitation driving datasets is a key factor in understanding soil moisture, and the choice of appropriate driving data is important to consider. This does not, however, provide any judgement

about the relative importance of soil moisture and precipitation datasets for understanding landslide susceptibility.

Figure 5 has too many lines and is difficult to read.

We have, for legibility, increased the size of this figure.

Figure 6 is not discussed in the main text and is only cited at L257. Is it useful?

We consider that Figure 6 provides useful context and an excellent illustration of the close fit between soil moisture data and the exponential function in some cases, despite this evaluation not being the most effective after comparison.

L275: As I understand, the Authors show the use of the best set of parameters from Site 3, applying them to Site 1 (Figure 8). Results from other combinations are not shown but are relevant. I would also expect to see a comparison to the KGE distribution based on the best ensemble member from Site 1. The legend of Figure 8 says that ensemble members were driven using CHIRPS, but plot (d) indicates MSWX driving data.

We have corrected the figure caption to correctly highlight that both CHIRPS and MSWX driving data were used here. We have selected this combination as these two precipitation datasets provide the best performance.

Reviewer 2

Thank you for your positive feedback on our manuscript. We appreciate your constructive comments and suggestions, which have helped us improve the clarity and significance of our work. Below, we address each of your points in detail.

The objective of this research is to explore the feasibility of using sparse field observations to calibrate the more accurate soil moisture and thus improve the accuracy of landslide forecasting in Nepal. It is of some significance for landslide susceptibility mapping in Nepal and other regions where field measurements of soil moisture are limited.

However, there are some errors/suggestions should be modified in the manuscript:

Could you please give the correlation between soil moisture and landslide susceptibility? Which is very important for this research. I recommend you add a section of "Application", in which, you can use your model to make a "landslide susceptibility mapping in the study area" to highlight your work's importance.

Thank you for emphasizing the importance of explicitly establishing the correlation between soil moisture and landslide susceptibility. Soil moisture affects landslide susceptibility by increasing soil density and weight, elevating porewater pressure (reducing soil strength), and adding surface water weight. Saturation can also prevent infiltration, causing runoff and erosion, further destabilizing slopes. These relations are well established from a variety of field and laboratory studies and from basic soil mechanics. We have expanded our description of this, while re-organizing our introduction, to better highlight this. By incorporating this information, we establish a clear connection between our soil moisture modeling efforts and their relevance to landsliding. We also aim to more clearly establish the bounds of this study, which does not aim to directly predict landsliding. As such, we do not add a dedicated application section but do discuss this in more detail throughout.

P13, Figure 6. Please give the meaning of figure a, b, c, d and e in the figure name.
P14, Figure 7. Please give the meaning of figure a, b, c, d and e in the figure name. Please also explain the meaning of different colors in the figure name.
    5. P15, Figure 8. Please give the meaning of figure a, b, c, d and e in the figure name. Please also explain the meaning of different colors in the figure name.

Thank you for pointing out the need for clearer figure captions. We have revised the caption for each of these figures to include detailed explanations of each subplot.

Overall, we have revised our manuscript to address the reviewers' concerns. Key changes include reorganizing the introduction and clarifying soil moisture's role as a landslide preconditioning factor. Our revisions, driven by the reviewer suggestions, strengthen the study's contribution to scaling sparse soil moisture data and improving landslide hazard understanding in data-limited regions. We hope that they also clarify our overall objectives with this manuscript and why we consider it a good fit to this journal. We once again thank the reviewers for their time.